# A Genetic-Algorithm-Based Approach for Optimizing Tool Utilization and Makespan in FMS Scheduling

Andrea Grassi [1,†] , Guido Guizzi [1,†] , Valentina Popolo [1,†] and Silvestro Vespoli [2,*]

1 Dipartimento di Ingegneria Chimica, dei Materiali e della Produzione Industriale (DICMAPI), Università degli Studi di Napoli Federico II, Piazzale Tecchio 80, 80125 Napoli, Italy; andrea.grassi@unina.it (A.G.); guido.guizzi@unina.it (G.G.); valentina.popolo@unina.it (V.P.)
2 Facoltà di Scienze Giuridiche ed Economiche, Università Telematica Pegaso, 00186 Rome, Italy
* Correspondence: silvestro.vespoli@unipegaso.it
† These authors contributed equally to this work.

**Abstract:** This paper proposes a genetic algorithm approach to solve the identical parallel machines problem with tooling constraints in job shop flexible manufacturing systems (JS-FMSs) with the consideration of tool wear. The approach takes into account the residual useful life of tools and allocates a set of jobs with specific processing times and tooling requirements on identical parallel machines. Two metrics are introduced to evaluate the scheduling decisions and optimize the scheduling process, with the competitive goal of maximizing tool utilization and minimizing production makespan. The proposed approach searches for a set of optimal solutions on the Pareto front that offers the best possible balance between these two objectives, achieving optimal local performance in terms of both makespan and tool utilization. The approach is implemented with a customized genetic algorithm and validated on a real case study from a company operating in the aerospace sector, which confirms its effectiveness in increasing tool utilization and reducing the makespan. The results show that the proposed approach has significant practical implications for the manufacturing industry, particularly in the production of high-value materials such as those in the aerospace sector that require costly tools. This paper contributes to the operational research community by providing advanced scheduling algorithms that can optimize both the makespan and the tool utilization concurrently, improving production efficiency and maintaining competitiveness in the manufacturing industry.

**Keywords:** flexible manufacturing systems; operations scheduling; optimization; tool utilization; makespan; genetic algorithm; aerospace case study

## 1. Introduction

In recent years, there has been a shift in the manufacturing industry towards more flexible and efficient production systems, and the flexible manufacturing system (FMS) has emerged as the leading solution. FMSs have grown in importance in the aerospace, automotive, and electronics industries as a result of their capacity to produce multiple products simultaneously and quickly adapt to production changes [1]. These systems are composed of interconnected workstations, automated material handling and storage systems, and an integrated computer system for control and coordination [2–4]. This enables FMSs to adapt to changing market demands and production requirements, ultimately leading to increased efficiency and competitiveness in the manufacturing industry [5].

FMSs can be divided into three main categories based on their job processing orders: open shops, flow shops, and job shops [6]. The job shop system is the most commonly used FMS due to its flexibility and adaptability, where each job is processed on available machines within a specified processing time, with the constraint that each machine can only process one operation per job. This system is known as the job shop flexible manufacturing system (JS-FMS), and it requires more complex scheduling algorithms to handle the flexibility and variety of jobs involved. This problem is known in the literature as the job shop scheduling

problem (JSP), and it seeks to assign production jobs to machines at particular times to optimize multiple objectives such as makespan, flow time, and tardiness [7,8]. As a classic problem in operational research, the JSP is known to be NP-hard, and this means that it is computationally difficult to solve, requiring advanced algorithms and techniques to find efficient solutions [9].

To address the complexity of scheduling JS-FMSs, researchers have proposed different optimization techniques to solve the JSP, both with exact methods (e.g., mixed-integer models) and approximate methods (e.g., simulation, neural networks, genetic algorithms, and simulated annealing) [10,11]. One critical issue in the scheduling of an FMS is tool deterioration, which has been widely addressed from the scientific literature [12–16]. For example, Hirvikorpi et al. [17] developed a genetic algorithm to solve the job scheduling with stochastic tool lifetime (JSSTL) problem and showed that the proposed algorithm outperformed the traditional short processing time (SPT) method. On the other hand, Xiuli et al. [18] proposed a multiobjective hybrid pigeon-inspired optimization and simulated annealing (MOHPIOSA) algorithm to tackle the FJSP by simultaneously considering the effects of tool deterioration and energy consumption. In recent work, Salama and Srinivas [19] proposed a similar sustainability-oriented approach to scheduling with tool deterioration in order to minimize the weighted costs of energy consumption, integrating the information about tool costs and production delays.

In addition to the scheduling challenges, the manufacturing industry also focuses on improving tool path optimization and machining processes to enhance production efficiency and reduce errors. For instance, a study by Sato and Yan [20] presented a method for optimizing the tool path for an independently controlled fast tool servo, aiming to reduce form errors in a single step of machining during freeform surface diamond turning. Moreover, there has been increasing interest in advanced lubrication techniques, such as electrostatic atomization minimum quantity lubrication (EA-MQL), to improve machining performance and reduce environmental impact. A comprehensive review by Xu et al. [21] discusses the mechanism and applications of EA-MQL machining, highlighting its benefits and challenges in various manufacturing contexts.

However, the analyzed approaches still do not adequately take into account the phenomenon of tool wear from an operational point of view. As a matter of fact, operators often change tools prematurely to avoid breaking them during a shift, resulting in suboptimal tool utilization and increased costs [22]. This is a significant problem, especially in the production of high-value materials such as those in the aerospace sector that require costly tools, where optimization of their utilization is of utmost importance to minimize production costs [23–25]. In the current literature, there are studies that propose scheduling algorithms with the objective of minimizing tool wear [26,27]. However, it has been observed that there remains a lack of research that simultaneously addresses both the need for preserving tool consumption and minimizing makespan in a comprehensive manner. Therefore, in this paper, we propose an innovative scheduling approach for JS-FMS that addresses this gap in the literature by concurrently optimizing both tool utilization and makespan, ensuring an effective balance between these competing objectives. This novel method will provide valuable insights for the manufacturing industry, particularly in the production of small-series products, where meeting customer demand and maintaining competitiveness are of utmost importance [28]. As a matter of fact, an optimal work sequence can improve tool utilization and reduce the number of partially used tools, making scheduling an FMS a challenge. The scientific contributions of this work are as follows:

- The introduction of a new method to model and allocate a set of jobs with specific processing times and tooling requirements on identical parallel machines, considering both the job and tool assignment based on the residual useful life of tools;
- The proposal of two novel metrics to evaluate scheduling decisions, aiming to optimize both tool utilization and production makespan;
- The effective balance of the competing objectives of tool utilization and makespan minimization by identifying a set of optimal solutions on the Pareto front.

By considering the impact of tool wear on scheduling optimization, this approach advances the state of the art in scheduling algorithms for JS-FMS. It provides a more accurate representation of real-world production scenarios, particularly in industries such as aerospace, where expensive tooling is required. The proposed approach was implemented with a customized genetic algorithm and validated on a real case study from a company located in Naples (Italy) and operating in the aerospace sector. The algorithm, as conceived, provides practitioners with quantitative insights about the optimal configuration of the FMS with respect to the management of the tool warehouse, whether it should be centralized or decentralized, also supporting the optimal scheduling process by both increasing tool utilization and makespan reduction in JS-FMSs.

The remainder of the paper is organized as follows. Section 2 describes the hypothesis on the problem under consideration; Section 3 introduces the proposed genetic algorithm architecture; Section 4 presents the experimental scenario and the discussion of the results; Section 5 concludes the paper.

## 2. Problem Formulation

The optimization of production scheduling for flexible manufacturing systems (FMSs) is a crucial task in industrial settings, especially in highly demanding industries such as aerospace, automotive, and electronics [29]. The efficient allocation of jobs to parallel machines and the management of tools are essential to ensure productivity, minimize costs, and maintain competitiveness [30]. As mentioned in Section 1, current approaches available in the literature do not adequately consider the phenomenon of tool wear, leading to suboptimal tool utilization, increased costs, and waste of tool residual life [14,22]. The problem is even more pressing when dealing with high-value materials, which require the use of costly tools [31]. Therefore, there is a clear need for advanced scheduling algorithms that can optimize both the makespan and the tool utilization concurrently, while taking into account the phenomenon of tool wear. Such an algorithm could potentially reduce costs, improve efficiency, and increase competitiveness for industries that rely on FMS.

The identical parallel machines problem with tooling constraints is the problem explored in this paper. The scenario involves different jobs, each requiring specific tools for machining. Processing time varies for each job and is not dependent on the machine it is performed on. Each machine has a tool warehouse with limited capacity and automatic tool changer, allowing it to process multiple jobs without significant setup times, as long as the required tools are distinct. A constraint of this problem is that each machine can only process one operation at a time. If a job requires multiple operations and different tools, these operations must be performed in sequence on the same machine. However, interrupting an operation is not feasible as the process cannot be resumed from its interruption point. The production system includes a double pallet that eliminates the wait for setup times between operations on the same machine.

The goal is to find the best possible sequencing of jobs allocated to different machines in order to (i) maximize the utilization of the tools' useful life and avoid having tools that remain with a residual useful life that cannot be used for next operations, and (ii) keep the makespan at the minimum possible with respect to the production plan. The proposed approach aims to solve this multiobjective optimization problem by minimizing the two target variables that measure the balancing of machines and the effectiveness of tool utilization. To achieve this, a measure of the two target variables and a genetic algorithm was developed. It can provide nondominated optimal solutions on the Pareto front, allowing for a better balance between the two proposed objectives.

The problem statement of this work is based on a real case of an aerospace industry company that produces titanium parts using FMSs for production. The company requires effective scheduling of production machines, particularly during unsupervised night shifts. Due to the high cost of tooling and the risk of tool breakage during machining, the company estimates the residual useful life of tools in a conservative manner, taking the advised value from the tool manufacturer. The problem they face is to optimally schedule jobs

among different machines in the FMS station, which all have independent automated tool warehouses. The proposed algorithm aims to identify the optimal configuration of the considered FMS with respect to the management of the tool warehouse, determining the tool to be loaded on each machine for dealing with the scheduled job.

### 3. The Proposed Approach

The proposed approach considers two target variables to optimize: the balancing of machines, the so-called "smoothness index" ($SX$), and the "effectiveness utilization tool" ($EUT$). The $SX$ is a traditional measure of the assembly line theory and represents a measure of the workload assigned to the various machines. It assumes a value of zero when production is perfectly balanced among FMS machines, and assumes the maximum value equal to the sum of all jobs' processing time when all machining time is concentrated on one machine. The $EUT$ is a dimensionless measure of how effective the job allocation is in the use of the tools; it assumes a value of zero for an ideal situation in which tools are not wasted, and assumes positive values when tool residual life is wasted. Therefore, we understand that $SX$ and $EUT$ are interrelated quantities. A solution that minimizes the value of $SX$ minimizes the umbalancing between the machines in terms of processing time (as shown in Figure 1), resulting in a lower makespan for the scheduled operations, but will result in higher tool waste due to suboptimal scheduling of the tools at the machines. On the other hand, a solution that minimizes the value of $EUT$ (as depicted in Figure 2) leads to more efficient tool utilization but creates a strong imbalance in the distribution of machining times across the machines, increasing the overall makespan value of the production system. This problem is a classic example of multiobjective optimization.

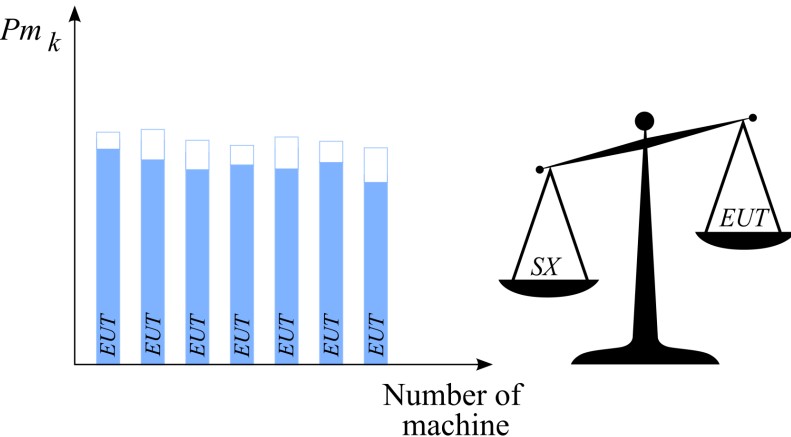

**Figure 1.** Machine load balance—solution minimizing $SX$.

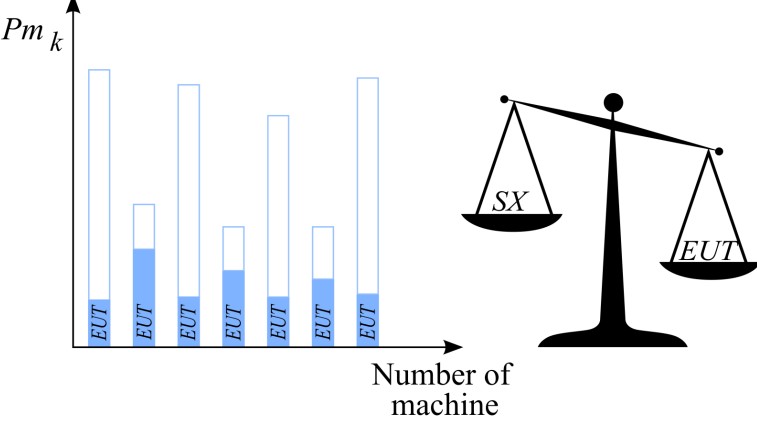

**Figure 2.** Tool utilization efficiency—solution minimizing $EUT$.

Using Graham notation ($\alpha \mid \beta \mid \gamma$), we can classify the problem considered in this paper as follows:

$$\alpha = P \tag{1}$$

$$\beta = \varnothing \tag{2}$$

$$\gamma = SX, EUT \tag{3}$$

Equation (1) indicates that the problem involves single-stage job scheduling with $m_c$ identical parallel machines; this means that each of the $m_c$ machines can process any job independently and simultaneously. Equation (2) means that the jobs do not have any characteristics specified by Graham (e.g., preemption is allowed, presence of limited resources, precedence relations between jobs, release dates, processing time has a lower and upper bound); Equation (3) indicates that the optimal criteria are the minimization of the unbalancing of processing times between machines ($SX$) and the efficient use of tools ($EUT$).

Let us introduce the following notation:

- $n$ is the number of jobs to be processed;
- $m_c$ is the number of parallel machines;
- $t$ is the number of different types of tools required to produce the job orders;
- $j_i$ is the $i$-th job, $i = 1, \dots, n$;
- $m_k$ is the $k$-th machine, $k = 1, \dots, m$;
- $J_{m_k}$ is the set of jobs assigned to the machine $m_k$;
- $T_v$ is the $v$-th type of tool, $v = 1, \dots, t$;
- $UL_v$ is the useful life of the $v$-th tool (in machining minutes);
- $RUL_v$ is the residual useful life of the $v$-th tool (in machining minutes);
- $h_{i,b}$ is the $b$-th tool required to process the job $i$;
- $h_{i,b} \in H_i \subset T$, where $H_i$ is the set of different types of tools required to produce the job $i$ and $T$ is the set of different types of tools required to produce all the jobs;
- $p_{H_i,i}$ is the machining time (in minutes) of the job $i$ using the set of tools $H_i$;
- $P_{m_k}$ is the total machining time of the jobs assigned to the machine $m_k$;
- $\overline{P}$ is the average machine processing time.

Given the introduced notation, it is possible to calculate $SX$ as in Equation (4) and $EUT$ as in Equation (5).

$$SX = \sqrt{\sum_{m_k} (P_{m_k} - \overline{P})^2} \qquad \forall k \in \{1 \dots m_c\} \tag{4}$$

$$EUT = \sum_v EUT_v \quad \text{where} \quad EUT_v = \sum_v BestUT_v - UT_v \quad \forall v \in \{1 \dots t\} \tag{5}$$

where $BestUT_v$ (Equation (6)) is the utilization of the $v$-th tool type in the best (ideal) solution.

$$P_{m_k} = \sum p_{H_i,i} \qquad \forall i \in J_{m_k} : \overline{P} = \frac{\sum_{k=1}^{m_c} P_{m_k}}{m_c}$$

$$BestUT_v = \sum_i \frac{p_{i,v}}{UL_v} \quad \forall i \in 1 \dots n \tag{6}$$

To address the multiobjective optimization problem presented in this paper, a genetic algorithm (GA) is proposed, which is capable of generating optimal solutions for the scheduling problem. The choice of using a genetic algorithm is motivated by its ability to efficiently explore the solution space, find optimal or near-optimal solutions, and handle multiobjective problems through the use of Pareto front analysis. Although the algorithm takes inspiration from the traditional structure of a GA, the proposed approach includes specific modifications to the chromosome representation, crossover, and mutation operations. These adaptations enable the generation of high-performing solutions that effectively

balance both tool utilization and makespan. As illustrated in Figure 3, the flowchart of the proposed genetic algorithm is presented. This algorithm differs from a classical GA as it was customized for the identification of dominant solutions, which are those that cannot be improved in any objective without degrading at least one other objective. This leads to the construction of a Pareto front, a set of nondominated solutions representing the trade-offs between the competing objectives. In contrast, a classical GA focuses on finding the optimal solution to the problem through simple fitness assessment. In the following sections, each of these components will be described in detail, and their utilization in the proposed algorithm will be discussed.

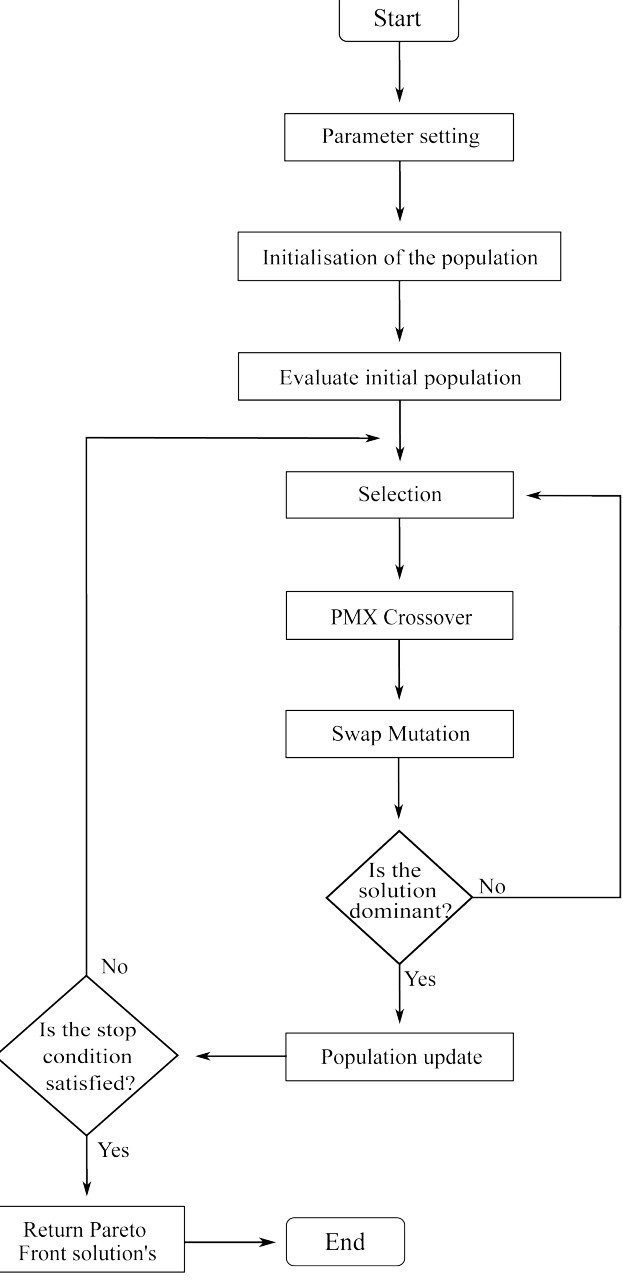

**Figure 3.** Flowchart of the proposed customized genetic algorithm for JS-FMS.

### 3.1. Chromosome

As the objective is to determine the optimal sequence of operations to be performed on various machines, it is imperative that the chromosome accurately represents this information. With this in mind, the chromosome was designed to represent the sequence

of operations scheduled on each machine. It is worth noting that the allocation of jobs to machines and the sequencing of those jobs on each machine are two important aspects of the scheduling problem. These aspects are captured in the chromosome through its positional encoding, where the position of each gene represents the machine to which the job has been assigned, and the sequencing of the job on that machine. The chromosome was designed with a fixed length, which is determined by the number of machines, the number of jobs, and the number of scheduling days considered in the problem. In the example shown in Figure 4, the chromosome was designed to allocate a maximum of four different jobs per day on the machines, and considers a total of two scheduling days. As such, the first four allocations of the chromosome represent the jobs assigned to the first machine on the first day, the next four represent the jobs assigned to the first machine on the second day, and so on. It is also worth mentioning that, once the chromosome is defined, its dimensionality cannot be changed during the execution of the algorithm. To account for this, the presence of zeros was taken into consideration in the chromosome design, allowing solutions to be identified even if not all possible allocation slots are occupied. In this context, the zeros are simply skipped, as shown in the example in Figure 4.

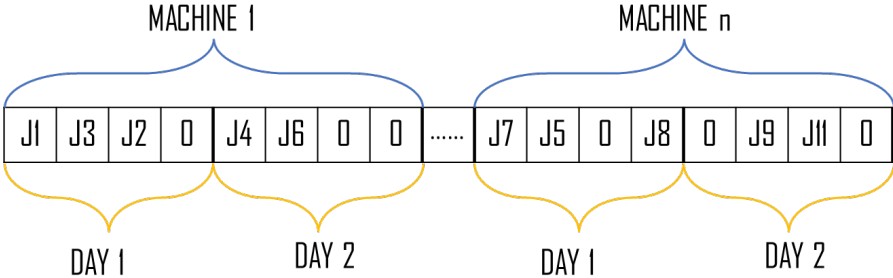

**Figure 4.** Example of chromosome representation in the genetic algorithm solution space.

### 3.2. Crossover Operation

The crossover operation in a genetic algorithm is the process of generating a child solution by combining the genetic information of two parent solutions. The purpose of this operation is to create offspring that are fitter and more diverse than their parents, thereby enriching the population with better individuals. The crossover operator is modeled after biological reproduction, where genetic information is passed from one generation to the next. In this study, the partially mapped crossover (PMX) method was adopted to generate the child chromosome. This method is advantageous as it preserves the order and interconnections within the chromosome and ensures that the offspring respects the rules of permutation.

The process of PMX starts with the random selection of two parent chromosomes (P1 and P2) and two crossover sites. As illustrated in Figure 5, the first parent (P1) segment between the two sites is directly copied to the same position of the second child (O1). Then, the elements that are present in the middle segment of the second parent (P2) but not in P1 (elements 1, 9, and 6 in the illustration) are placed in the corresponding positions of the child chromosome. For instance, element 9 in P2 is positioned at 5 in O1, so the next step is to place element 9 in the available position from the previous 5 in P2. This process continues for elements 6 and 1 in a similar manner. Finally, the remaining elements of parent P2 are copied to the corresponding positions of the child chromosome. This approach ensures that the offspring chromosome inherits traits from both parents while maintaining the order and interconnections of the solution. By doing so, the PMX method helps maintain the diversity of the population and improves the chances of finding an optimal solution.

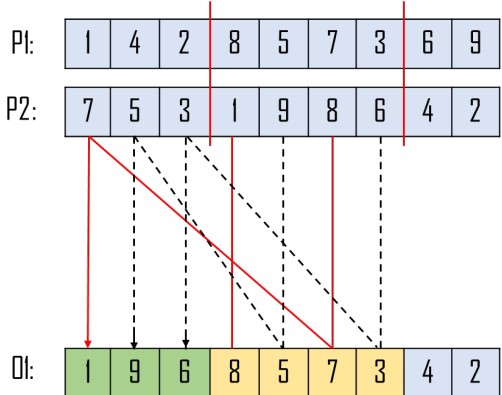

**Figure 5.** Example of partially mapped crossover (PMX) operation between two parent chromosomes to generate a child chromosome.

### 3.3. Mutation Operation

The purpose of mutation in genetic algorithms is to introduce new genetic information into the population, breaking away from the constraints imposed by the current solutions. This helps the algorithm escape from being trapped in a local minimum and aids in exploring the entire search space. Mutation is crucial in maintaining the genetic diversity of the population, thereby increasing the chances of discovering better solutions. In the present work, random resetting is used as the mutation method. This method is equivalent to binary mutation, where each gene has a fixed probability, $p_m$, of being replaced by a random value, calculated within a predetermined range. This approach ensures that the mutation rate is independent for each gene, allowing for a more nuanced exploration of the search space.

## 4. Results and Discussion

The proposed approach for the identical parallel machines problem with tooling constraints was tested and validated in a real-world case study of a manufacturing company in the aeronautical supply chain. As previously stated, the company must schedule production machines during unsupervised shifts while efficiently using costly tools with conservative estimates of their residual useful life. To assess the approach's ability to optimize machine balancing and tool utilization, a genetic algorithm was implemented in Python, taking into account the chromosome configuration and genetic operators described earlier. The experimental methodology was designed to evaluate the approach's effectiveness, with levels of the considered factors based on their relevance to the company's real-world scenario.

In Section 4.1, the design used to evaluate the proposed approach and the performance measures used to assess the solutions' quality are explained, along with the rationale behind the selection of factors and their levels. In Section 4.2, instead, the results of the experiments are analyzed, and implications for practitioners are discussed. The discussion section highlights the importance of considering tool residual life when scheduling production machines and the potential impact of the proposed approach in reducing tool-related costs and improving tool utilization in similar real-world scenarios. Additionally, the results suggest that the optimal configuration of the FMS tool warehouse, whether centralized or decentralized, may vary depending on the specific scenario being considered.

### 4.1. Experimental Methodology

The experimental methodology aims to evaluate the proposed approach by testing it on a simulated scenario inspired by a real-world scenario from the aerospace industry. The company's production system is made up of fully autonomous FMS units, equipped with an internal tool warehouse. These machining operations require efficient scheduling, and the *SX* and *EUT* parameters play a crucial role in understanding the type of solution

identified by the proposed approach. Minimizing the $SX$ parameter results in a solution with the lowest makespan, where each of the plant's productive FMS units has an equal distribution of work. This represents the fastest solution to complete the scheduling, but with the use of tools dispersed and replicated among the various machining units, leading to higher $EUT$ values. On the other hand, solutions that minimize the $EUT$ value result in a situation where some machining units are occupied for much longer than others, with a unbalanced load between machines and a longer overall makespan, saving the waste of tool life.

The Python 3.10.4 version was used for implementing the genetic algorithm in this study. This choice was made as it is one of the most recent and stable versions of Python, offering improved performance, enhanced features, and better library support, which ensures reliable and efficient execution of the algorithm. To validate the proposed algorithm, production scenarios of 200 total jobs composed of 1400 operations were generated. The operation processing times were extracted from a triangular distribution with a minimum value of 15 min, a maximum value of 391, and a modal value of 98. The experimental scenarios were generated varying two factors: the number of different tools used in the machining cycles and the distribution of the different types of tools in the machining cycles. Table 1 shows the three distinct values for the first factor, with the central value being representative of the case study. These values simulate scenarios in which 56, 75, and 94 tool types are used in the machining cycles. Three levels were also determined for the second factor, with the central value always representative of the case study. The three values represent the frequency distribution of the specific tool type within the generated processing cycles. A value of 00 is representative of a situation in which the use of tools within the machining cycles is uniform, meaning that the generated operation's technological cycles present shared tools among them. Figure 6 depicts the frequency distribution probability, where the type of tool is represented on the x-axis and the frequency distribution on the y-axis. On the other hand, both the 03 and 06 values represent a damped exponential tool frequency distribution. However, the 03 value indicates a less pronounced damped exponential distribution, meaning that it is closer to a uniform situation than the distribution represented by the 06 value, as shown in Figures 7 and 8.

**Table 1.** Factorial scenario plan.

| Experimental Factor | Levels | Unit |
|---|---|---|
| Machines | 3 | [machine] |
| Job | 200 | [job] |
| Total operation | 1400 | [operation] |
| Operation time distribution | triangular $(15, 391, 98)$ | [minutes] |
| Type of tool | 56–75–94 | [different tool] |
| Tool type distribution scenario | 00–03–06 | [scenario] |

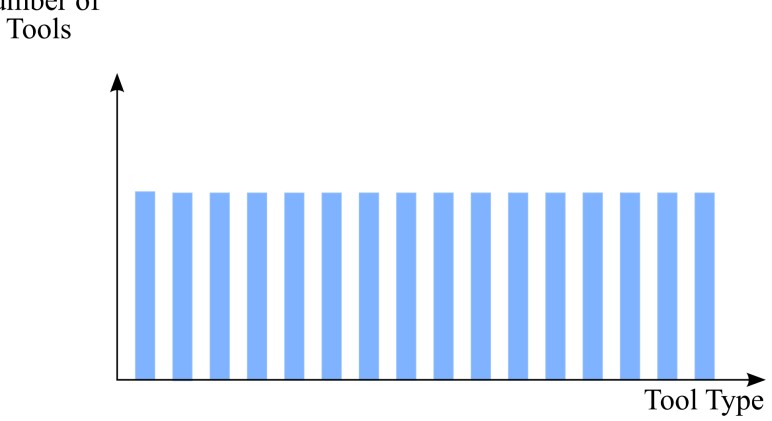

**Figure 6.** Uniform tool utilization distribution—Tool Type Distribution Scenario 00.

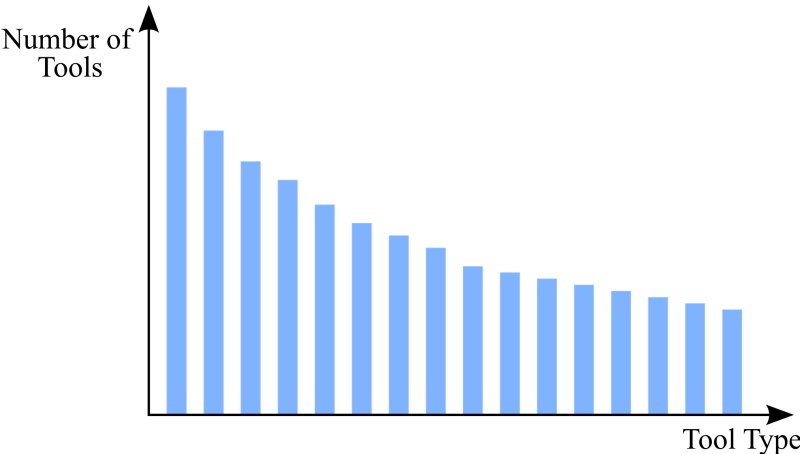

**Figure 7.** Frequency probability distribution of tool types with a damped exponential distribution—Tool Type Distribution Scenario 03.

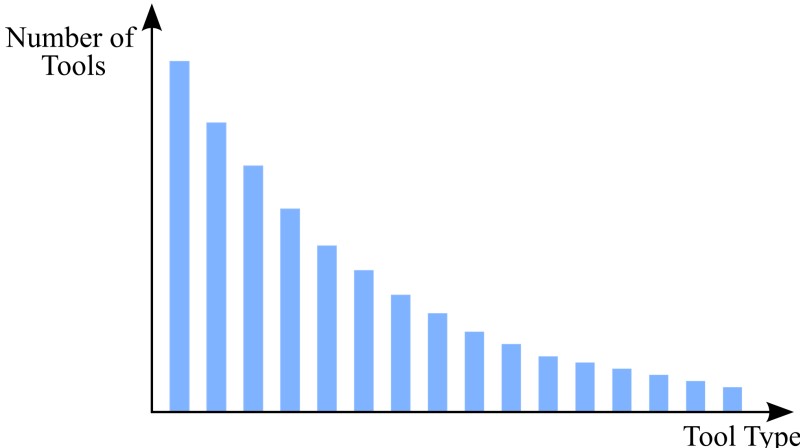

**Figure 8.** Frequency probability distribution of tool types with a more pronounced damped exponential distribution—Tool Type Distribution Scenario 06.

To ensure the robustness of the proposed approach, the experimental methodology involved generating a sufficient number of problems for each combination of factors. For the proposed factor and level, with a full factorial experimental plan, nine different scenario were identified. Specifically, a generation problem algorithm was built for generating 10 different problems for each combination, resulting in a total of 90 runs. To evaluate the proposed approach, the genetic algorithm discussed in Section 3 was applied to each of the resulting scenarios, and the solutions were analyzed to gain insight into the algorithm's performance in terms of distribution and resilience.

### 4.2. Results and Discussion

In this section, we present the most relevant results obtained from our experiment. The results are shown through scatter plots, with the x-axis representing the value of the objective function *EUT* and the y-axis representing the value of the objective function *SX*. As a reminder, the goal of the proposed genetic algorithm is to minimize the weighted sum of these two objective functions, obtaining a Pareto front of optimal solutions that allow the decision-maker to choose from among them the appropriate solution depending on the situation at hand. The results will be presented in two steps: first, the results obtained by keeping the distribution of the tool type constant while allowing the number of tools to vary; second, the results obtained by keeping the number of tools constant while observing what happens when the distribution of the tool type varies.

Figures 9 and 10 depict the first three distinct scenarios, respectively: Figure 9 represents scenarios where the tool type distribution in technological cycles is uniform; Figure 11 represents scenarios with the imbalance shown in Figure 7; and Figure 10 represents scenarios with the imbalance shown in Figure 8. In these figures, the colors blue, red, and green represent different classes of solutions corresponding to scenarios with a number of 56, 75, and 96 tools, respectively. The results indicate that as the number of tool types increases, with the same distribution of tools among the technological cycles, the Pareto front rises and the slope of the front increases. This trend is repeated in all the various distribution scenarios analyzed, although it should be noted that this impact is stronger in the situation with uniform tool distribution and gradually decreases in the situations with increasing imbalance. This means that with the same $SX$, the respective $EUT$ value increases, making it more difficult to optimize tool utilization as the number of tools increases. Conversely, as the $EUT$ remains the same, the imbalance between the machines increases significantly (the $SX$ value identified by the optimal solution increases).

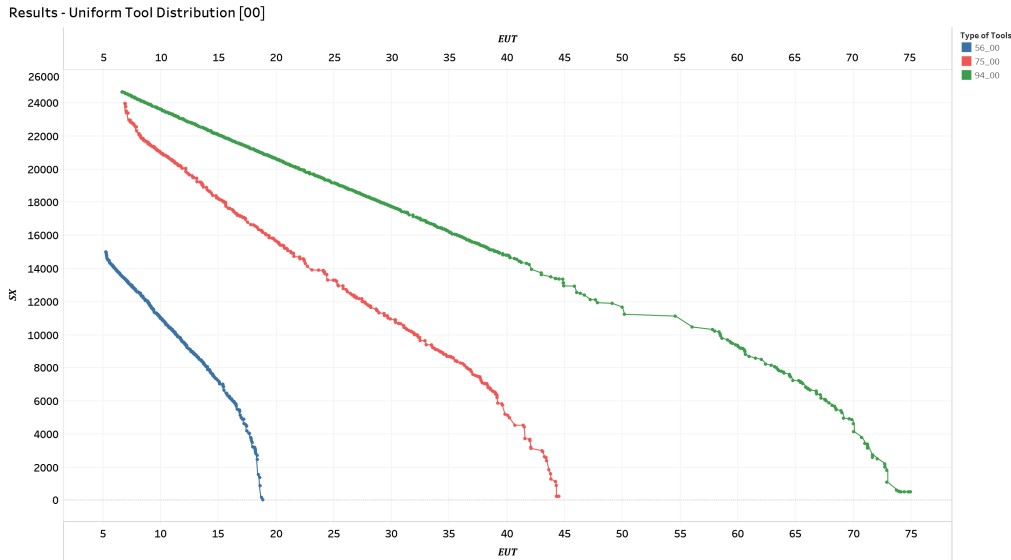

**Figure 9.** Domain solution—Uniform Tool Distribution [00]. Blue, red, and green colors represent solutions corresponding to scenarios with 56, 75, and 96 tools, respectively.

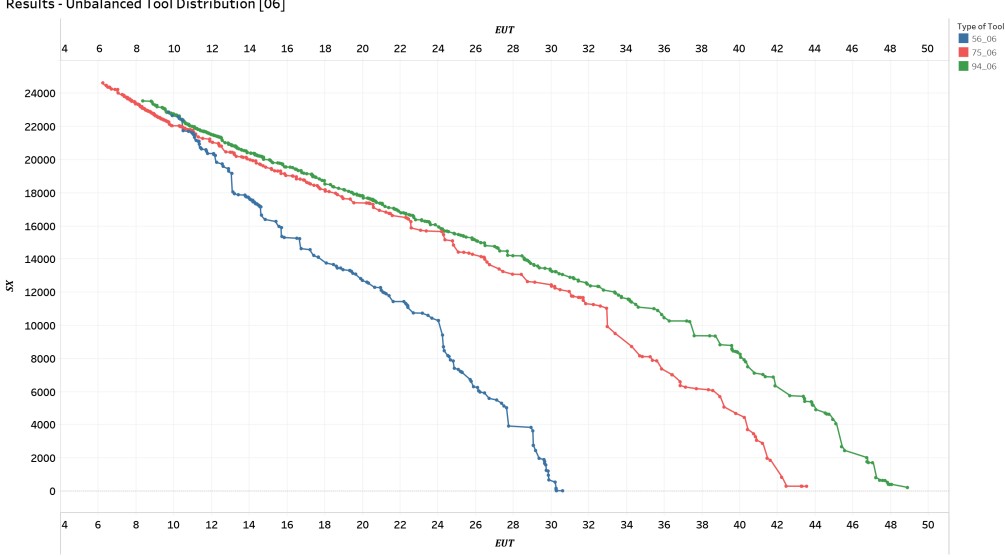

**Figure 10.** Domain Solution—Unbalanced Tool Distribution [06]. Blue, red, and green colors represent solutions corresponding to scenarios with 56, 75, and 96 tools, respectively.

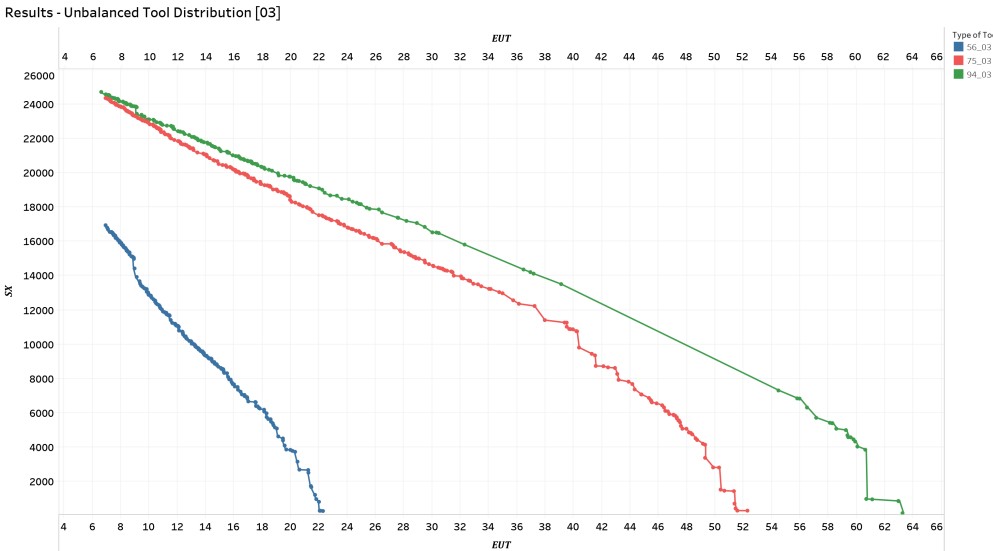

**Figure 11.** Domain solution—Unbalanced Tool Distribution [03]. Blue, red, and green colors represent solutions corresponding to scenarios with 56, 75, and 96 tools, respectively.

From these results, we can highlight a practical conclusion: as the number of tool types required for machining operations increases, it becomes increasingly complex to optimize machining cycles while minimizing the makespan and safeguarding the tool life wastage. In such scenarios, it may be more convenient to organize the FMS unit to use a central tool warehouse rather than a decentralized onboard machine warehouse, to allow for combined optimizations with respect to both the makespan ($SX$) and the residual useful life of the tools ($EUT$). This consideration is not necessary if the distribution of tools in the machining cycles is uneven, as the gain from centralizing the tool warehouse becomes significantly reduced with increasing imbalance and the number of tools.

Finally, we focus on the results obtained when the number of tool types is fixed and the distribution of tool types varies. Figures 12–14 depict, respectively, the three distinct scenarios: Figure 12 depicts a scenario in which the number of tool types is at its lowest value (56), Figure 13 depicts a scenario in which the number of tool types is 75, and Figure 14 depicts a scenario in which the number tool types is 94. In these figures, the colors light teal, brown, and red represent different classes of solutions corresponding to scenarios with Uniform [00], Unbalanced [03], and Unbalanced [06] distribution, respectively. The results show that, with a fixed number of possible tools to be used, as the distribution of the tools between the technological cycles changes, better results are obtained in scenarios with a large number of tools and a uniform distribution between the machines (i.e., solutions with decentralized tool warehouse). This consideration becomes less important as the number of different tools increases and reverses in scenarios involving 94 distinct types of tools. On a practical level, we can conclude that if a small number of tools are predominantly used in the technological cycles, it is advantageous to use an onboardmachine warehouse, while a centralized warehouse is more advantageous in the case where the number of tools is large.

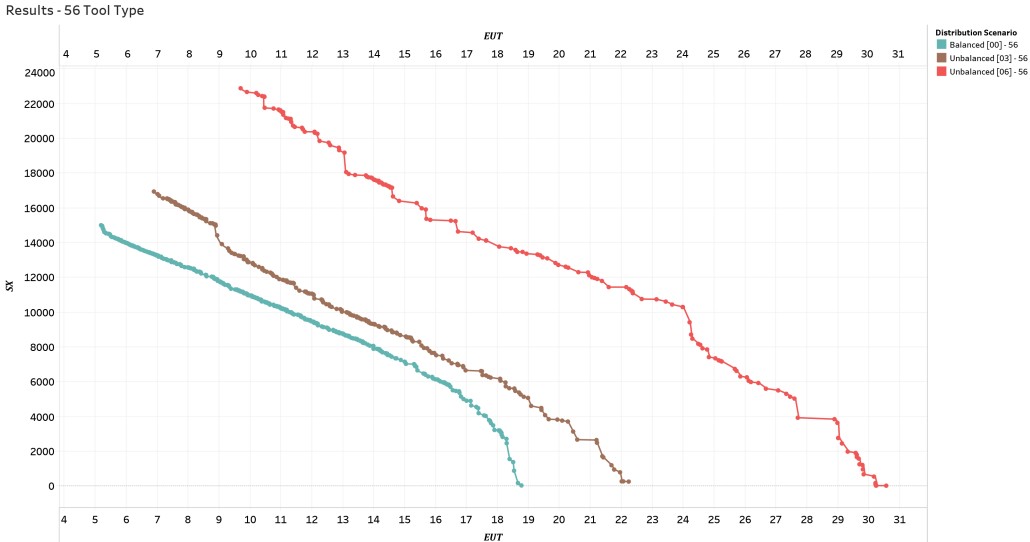

**Figure 12.** Domain Solution—fixed tool type at 56. Light teal, brown, and red colors represent solutions corresponding to scenarios with Uniform [00], Unbalanced [03], and Unbalanced [06] distribution, respectively.

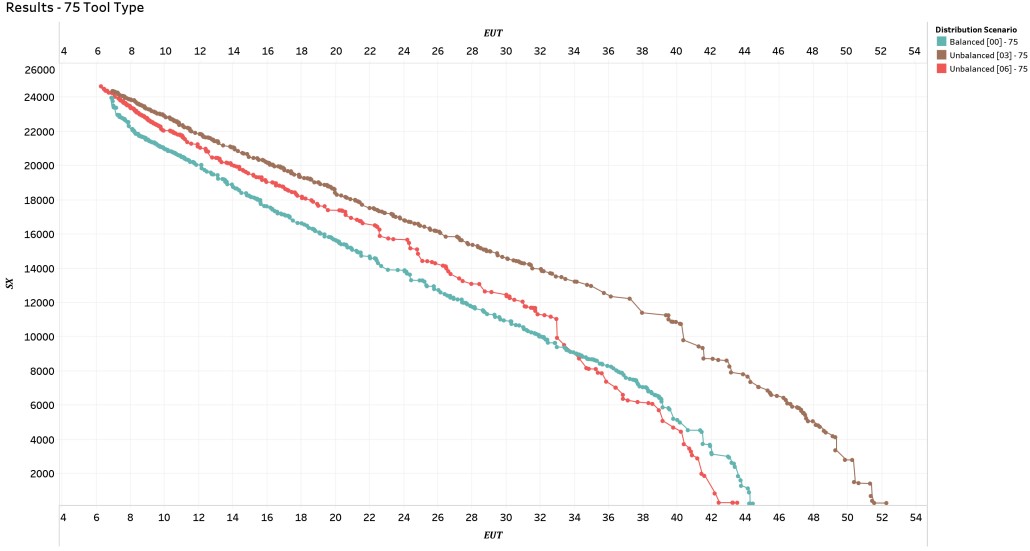

**Figure 13.** Domain Solution—fixed tool type at 75. Light teal, brown, and red colors represent solutions corresponding to scenarios with Uniform [00], Unbalanced [03], and Unbalanced [06] distribution, respectively.

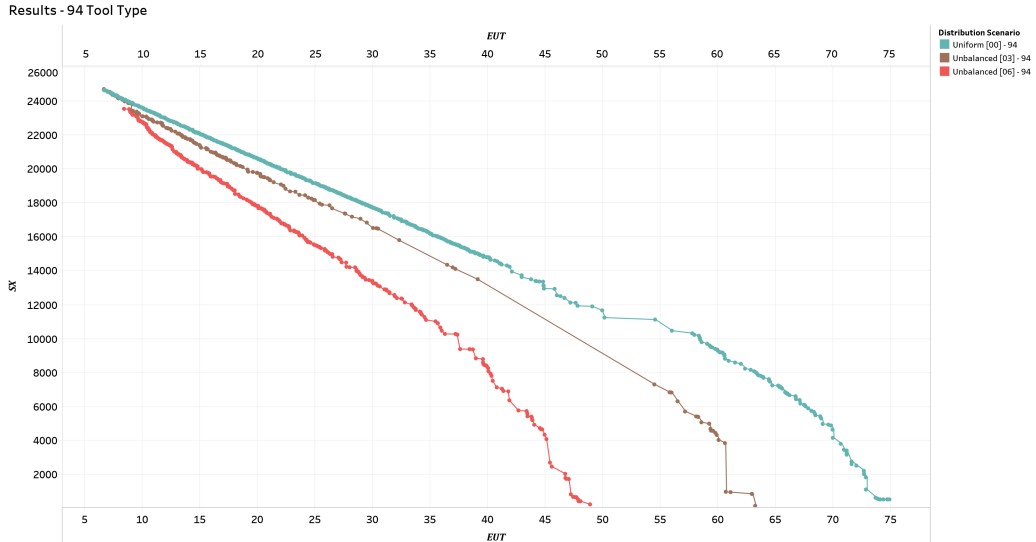

**Figure 14.** Domain Solution—fixed tool type at 94. Light teal, brown, and red colors represent solutions corresponding to scenarios with Uniform [00], Unbalanced [03], and Unbalanced [06] distribution, respectively.

## 5. Conclusions

In this paper, a novel approach to address the job shop scheduling problem in the context of job shop flexible manufacturing systems with the consideration of tool wear was presented. The proposed approach takes into account the residual useful life of tools conservatively estimated by manufacturers and allocates a set of jobs with specific processing times and tooling requirements on identical parallel machines. We introduced two metrics to evaluate the scheduling decisions and optimize the scheduling process, with the goal of maximizing tool utilization and minimizing production makespan. To address the trade-off between these two objectives, the proposed approach searches for a set of optimal solutions on the Pareto front that offers the best possible balance between them, achieving optimal local performance in terms of both makespan and tool utilization. We implemented this approach with a customized genetic algorithm and validated it on a real case study from a company operating in the aerospace sector, which confirmed the effectiveness of the approach in increasing tool utilization and reducing the makespan.

The results obtained from the considered experiment show that, as the number of tool types increases, it becomes increasingly complex to optimize machining cycles while minimizing the makespan and safeguarding tool life wastage. In such scenarios, it may be more convenient to organize the FMS unit to use a central tool warehouse rather than a decentralized onboard machine warehouse to allow for combined optimizations with respect to both the makespan and the residual useful life of the tools. However, if the distribution of tools in the machining cycles is uneven, the gain from centralizing the tool warehouse becomes significantly reduced with increasing imbalance and the number of tools.

The proposed approach has significant practical implications for the manufacturing industry, particularly in the production of high-value materials such as those in the aerospace sector that require costly tools. By optimizing tool utilization, the proposed approach can help reduce production costs, improve production efficiency, and maintain competitiveness. Additionally, in the production of small-series products, the proposed approach can help meet customer demand by reducing the makespan while improving tool utilization. Moreover, the solutions found by the proposed algorithm can be chosen by the production manager by selecting the solution that best satisfies the contingent requirement of the moment, for example, by choosing the one with the lowest makespan in hectic contexts or the others that preserve tool useful life waste.

Future research may focus on extending the proposed approach to include more complex scheduling scenarios, such as considering the stochastic tool life and the uncertainty of processing times. Moreover, combining the proposed approach with other optimization techniques may lead to more advanced algorithms and better performance. Finally, applying the proposed approach to other manufacturing sectors and scenarios may provide further insights into the effectiveness and efficiency of the approach.

**Author Contributions:** Conceptualization, A.G., G.G., V.P. and S.V.; Methodology, A.G., G.G., V.P. and S.V.; Software, A.G. and G.G.; Validation, G.G.; Formal analysis, S.V. and A.G.; Investigation, G.G. and A.G.; Resources, G.G.; Data curation, G.G. and V.P.; Writing—original draft, S.V. and V.P.; Writing—review & editing, S.V. and V.P.; Supervision, A.G., G.G., V.P. and S.V. All authors have read and agreed to the published version of the manuscript.

**Funding:** This research was funded by Tecnologica Srl—MOM 4.0 Project (CUP: B31B20000390005).

**Institutional Review Board Statement:** Not applicable.

**Informed Consent Statement:** Not applicable.

**Data Availability Statement:** Not applicable.

**Conflicts of Interest:** The authors declare no conflict of interest.

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
