# Peer review of "A Genetic-Algorithm-Based Approach for Optimizing Tool Utilization and Makespan in FMS Scheduling"

_jmmp, doi:10.3390/jmmp7020075_

Round 1

Reviewer 1 Report

This paper proposes a genetic algorithm approach to solve the Identical Parallel Machines Problem with Tooling Constraints in Job Shop Flexible Manufacturing Systems with the consideration of tool wear. The approach takes into account the residual useful life of tools and allocates a set of jobs with specific processing times and tooling requirements on identical parallel machines. This paper is rich in content, but it needs a major revision according to the following comments:

1. The effect of the proposed algorithm in the actual scene lacks an intuitive evaluation index;

2. In the citation part, the author should indicate whether there is any existing research similar to the research content of this paper, and reflect the innovation and necessity of this research through comparison;

3. It is recommended that the author provide the pseudocode of the genetic algorithm so that other researchers can reproduce the research content of this paper;

4. The content of the pictures in the paper is not clear, the necessary title is missing, and the text content of some pictures is not clear enough;

5. The following two papers, the first is about the method for optimizing tool path for the independent control fast tool servo to reduce form errors in a single step of machining, the second is a comprehensive review and critical assessment of the existing understanding of electrostatic atomization minimum quantity lubrication, which may be helpful for the introduction part of this paper.

[1] Tool path generation and optimization for freeform surface diamond turning based on an independently controlled fast tool servo. Int. J. Extrem. Manuf. 4 022002.

[2] Electrostatic atomization minimum quantity lubrication machining: from mechanism to application. Int. J. Extrem. Manuf. 4 042003 (2022).

Author Response

First, we would like to express our gratitude to the Editor for providing us with the opportunity to resubmit a revised version of the paper. We also wish to extend our appreciation to the Reviewers for their thorough examination of the manuscript and for their valuable comments, which have allowed us to significantly enhance the original version of the paper. We have diligently addressed each observation raised by the Reviewers, and the manuscript has been amended accordingly.

Modifications made to the text in response to the Reviewers' comments and suggestions are clearly indicated in the manuscript, with changes to the English form highlighted in red, and significant modifications or additions to the content marked in blue. Below, we provide our response to each specific comment brought to our attention. We hope that we have successfully addressed all concerns and that the paper is now suitable for publication in Journal of Manufacturing and Materials Processing.

Comments No 1: […] The effect of the proposed algorithm in the actual scene lacks an intuitive evaluation index”

Our response: Thank you for your comment. We apologise for not being clear in this regard. The approach proposed in this paper aims to identify a class of dominant solutions on a Pareto front. Therefore, the innovative aspect of the work is the fact that, unlike existing literature, we do not identify the best possible solution for tool consumption optimisation or the best possible solution for minimising the makespan but rather a class of solutions through which the decision-maker can choose their preferred option. We have clarified this aspect, appropriately identifying the scientific contribution of the paper at the end of the Introduction. We have highlighted the revision in blue.

Comments No 2: “In the citation part, the author should indicate whether there is any existing research similar to the research content of this paper, and reflect the innovation and necessity of this research through comparison”

Our response: Thank you for this comment, and we apologise for not making this point evident in the Introduction of the paper. In conjunction with the comment above, we have included citations of similar scientific works, outlining the scientific contribution of the present work compared to them. The main contribution lies in the fact that we propose a multi-criteria algorithm instead of a single-criterion solution. Also in this case, the changes are identifiable in blue.

Comments No 3: It is recommended that the author provide the pseudocode of the genetic algorithm so that other researchers can reproduce the research content of this paper”

Our response: We appreciate your comment. In this regard, we have provided more information on the differences between the proposed approach and a classic genetic algorithm, presenting a flowchart (Figure 4) of the logical steps to allow for the reproducibility of the work.

Comments No 4: “The content of the pictures in the paper is not clear, the necessary title is missing, and the text content of some pictures is not clear enough”

Our response: Thank you very much for this comment. We apologise for the quality of the results presentation. Following this comment, we have thoroughly revised Figures 8 to 13 to make them consistent in colours and more visible in the results display. We have also revised both the text and the image captions.

Comments No 5: “The following two papers, the first is about the method for optimizing tool path for the independent control fast tool servo to reduce form errors in a single step of machining, the second is a comprehensive review and critical assessment of the existing understanding of electrostatic atomization minimum quantity lubrication, which may be helpful for the introduction part of this paper. […]”

Our response: We thank you for bringing these scientific papers to our attention. After analysing them, we have expanded the discussion in the Introduction to integrate them into the discourse. In this case, the changes have been highlighted in blue.

Reviewer 2 Report

Manuscript Number: jmmp-2303427-peer-review-v1

Title: Optimizing Tool Utilization and Makespan in FMS Scheduling: A Genetic Algorithm Approach

The focus of the study is on a genetic algorithm approach to solve the Identical Parallel Machines Problem with Tooling Constraints in Job Shop Flexible Manufacturing Systems (JS-FMSs). Two metrics are introduced to evaluate the scheduling decisions and optimize the scheduling process, with the competitive goal of maximizing tool utilization and minimizing makespan. The proposed approach searches for a set of optimal solutions on the Pareto front.

There is novelty in this paper. However, it requires a revision. Please see my comments below.

- My main issue is about the genetic algorithm (GA) used in this paper to solve the JS-FMS. Is it a version of GA that is customized to solve JS-FMS? More specifically, I wonder if any of its operators including crossover operation and mutation operation has significantly changed to obtain better results after solving the JS-FMS? If not, then it means that authors only employed a standard version of GA to solve the problem. Under this condition, the GA algorithm should be removed from the title of the paper since it is not the main contribution of authors and they mostly studied about the tool utilization.

- Using well-known Graham notation (α|β|γ), authors classified the problem considered in the paper. However, this classification does not show if the type of the problem is single machine, flow shop, job shop. Using P for the first segment is not enough. It should show the type of the problem as well.

This is very common to show the type of the problem in "α" segment of the classification. Please correct it to have a better problem statement.

- Some Figures have low resolution. For example, Figures 8-13. Please increase the resolution of these figures, make their size smaller, and group them.  

- Authors have developed Python code for the problem in this paper. Please say what is the version of the Python used to code the problem.

- Figure 8-13 shows results of the study. If you ask me, I believe that authors have not mentioned what the meaning of colors is. There is one short line as the caption of each figure, but it is not enough. Are colors showing details about "group-_exp"? Then say what are each of green, orange and brown color graphs.

- Authors should not use both British and American spelling in one article. Please do not mix the two in a single piece of writing. I have observed both of them in the paper like: utilise, utilize, behaviour, behavior, organise, organize, summarise, summarize, recognise, recognize, centralise, centralize, characterise, characterize, minimise, minimize, optimise, optimize.

- Never use etc. at the end of a series that begins with for example, e.g., including, such as, and the like, because these terms make etc. redundant: they already imply that the writer could offer other examples.

Page 2: such as makespan, flow time, tardiness, etc.

Page 2: e.g., simulation, neural networks, genetic algorithms, simulated annealing, etc.

-I can see inconsistency in this paper. For example, both "Figure" and "Fig." are used in the body of the paper. Please only use one of them, not both. Examples are “as shown in Figure 1”, “as depicted in Fig. 2”, “shown in Figure 3” and “in the example in Fig. 3”.

- I suggest citing the followings on FMSs in aerospace, automotive, and electronics industries in the first paragraph (optional): [a] Enabling flexible manufacturing system through the application of industry 4.0 technologies, Internet of Things and Cyber-Physical Systems, vol.2, pp. 49-62 [b] Stochastic optimization of two-machine flow shop robotic cells with controllable inspection times: From theory toward practice, Robotics and Computer Integrated Manufacturing, Vol. 61, pp. 101822

-Please avoid using abbreviation in the keywords list:

Flexible Manufacturing Systems (FMS) --> Flexible Manufacturing Systems

- Other errors:

Page 2: widely addessed from --> widely addressed from

Author Response

First, we would like to express our gratitude to the Editor for providing us with the opportunity to resubmit a revised version of the paper. We also wish to extend our appreciation to the Reviewers for their thorough examination of the manuscript and for their valuable comments, which have allowed us to significantly enhance the original version of the paper. We have diligently addressed each observation raised by the Reviewers, and the manuscript has been amended accordingly.

Modifications made to the text in response to the Reviewers' comments and suggestions are clearly indicated in the manuscript, with changes to the English form highlighted in red, and significant modifications or additions to the content marked in blue. Below, we provide our response to each specific comment brought to our attention. We hope that we have successfully addressed all concerns and that the paper is now suitable for publication in Journal of Manufacturing and Materials Processing.

Comments No 1: [..] My main issue is about the genetic algorithm (GA) used in this paper to solve the JS-FMS. Is it a version of GA that is customized to solve JS-FMS? More specifically, I wonder if any of its operators including crossover operation and mutation operation has significantly changed to obtain better results after solving the JS-FMS? If not, then it means that authors only employed a standard version of GA to solve the problem. Under this condition, the GA algorithm should be removed from the title of the paper since it is not the main contribution of authors and they mostly studied about the tool utilization”

Our response: We apologise for not being clear on this point and appreciate your comment. In this regard, we want to clarify that there are some differences compared to a classic genetic algorithm. We have discussed these differences in the third section, also introducing a flowchart (Figure 4) of the algorithm. The changes have been highlighted in the paper in blue. Additionally, taking your suggestion on board, we have also modified the paper's title to more coherently reflect that it is based on a genetic algorithm.

Comments No 2: “Using well-known Graham notation (α|β|γ), authors classified the problem considered in the paper. However, this classification does not show if the type of the problem is single machine, flow shop, job shop. Using P for the first segment is not enough. It should show the type of the problem as well. This is very common to show the type of the problem in "α" segment of the classification. Please correct it to have a better problem statement.”

Our response: Thank you very much for your comment. We apologise for not being clear in defining the problem. Upon revising the problem statement, we would like to clarify that it is a single-stage problem with parallel machines, which is correctly identified with the letter "P" in the alpha segment. Nonetheless, we have revised and further clarified this aspect in the textual explanation of the notation. You can find this revision highlighted in blue.

Comments No 3: Some Figures have low resolution. For example, Figures 8-13. Please increase the resolution of these figures, make their size smaller, and group them”

Our response: Thank you very much for your comment. Following your feedback, we have substantially revised Figures 8 to 13, proposing greater colour uniformity and revisiting their size as per your advice. We hope that they are now more readable, and the results displayed are clearer.

Comments No 4: Authors have developed Python code for the problem in this paper. Please say what is the version of the Python used to code the problem”

Our response: Thank you very much for this point. Indeed, we apologise for not discussing this aspect. The version used was 3.10.4. We have added this specification to the paper in Section 4. You can find it highlighted in blue.

Comments No 5: Figure 8-13 shows results of the study. If you ask me, I believe that authors have not mentioned what the meaning of colors is. There is one short line as the caption of each figure, but it is not enough. Are colors showing details about "group-_exp"? Then say what are each of green, orange and brown color graphs”

Our response: Thank you very much for your comment. We genuinely apologise for not clarifying the different colours within the text. On this point, we have revised the colours of the images so that they are more consistent with each other. We have also clarified both the text and the caption of each image regarding the correspondence between the colour and the experiment shown. We hope that the images are now clearer and the results more evident.

Comments No 6 Authors should not use both British and American spelling in one article. Please do not mix the two in a single piece of writing. I have observed both of them in the paper like: utilise, utilize, behaviour, behavior, organise, organize, summarise, summarize, recognise, recognize, centralise, centralize, characterise, characterize, minimise, minimize, optimise, optimize.”

Comments No 7 Never use etc. at the end of a series that begins with for example, e.g., including, such as, and the like, because these terms make etc. redundant: they already imply that the writer could offer other examples. Page 2: such as makespan, flow time, tardiness, etc.  Page 2: e.g., simulation, neural networks, genetic algorithms, simulated annealing, etc..”

Comments No 8 I can see inconsistency in this paper. For example, both "Figure" and "Fig." are used in the body of the paper. Please only use one of them, not both. Examples are “as shown in Figure 1”, “as depicted in Fig. 2”, “shown in Figure 3” and “in the example in Fig. 3”

Our response: Thank you very much for pointing out these inconsistencies. We sincerely apologise for not maintaining a consistent style. We have conducted a thorough revision on this point, standardising the entire discussion in British style and revising all the reported inconsistencies. The most important changes regarding the English language have been indicated in red.

Comments No 9: I suggest citing the followings on FMSs in aerospace, automotive, and electronics industries in the first paragraph (optional): [a] Enabling flexible manufacturing system through the application of industry 4.0 technologies, Internet of Things and Cyber-Physical Systems, vol.2, pp. 49-62 [b] Stochastic optimization of two-machine flow shop robotic cells with controllable inspection times: From theory toward practice, Robotics and Computer Integrated Manufacturing, Vol. 61, pp. 101822.”

Our response: Thank you very much for suggesting these two studies. After a thorough analysis of them, they have been integrated into the discussion in the introduction.

Comments No 10: Please avoid using abbreviation in the keywords list: Flexible Manufacturing Systems (FMS) --> Flexible Manufacturing Systems”

Comments No 11: Other errors: Page 2: widely addessed from --> widely addressed from”

Our response: Thank you very much for these comments. We confirm that we have amended the requested points.

Reviewer 3 Report

The present paper proposes a genetic algorithm based approach to solve the Identical Parallel Machines with Tooling Constraints Problem in Flexible Manufacturing Systems (JS-FMS) considering tool wear. This approach considers the residual lifetime of the tools and allocates a set of jobs with specific processing times and tool requirements on identical parallel machines. The authors introduce two metrics to evaluate scheduling decisions and optimize the scheduling process with the competing objective of maximizing tool utilization and minimizing production. The proposed approach searches for a set of optimal solutions on the Pareto front that offer the best possible balance between these two objectives, achieving optimal local performance in terms of tool production and utilization. The approach described in the paper has significant practical implications for the manufacturing industry, especially in the production of high-value materials such as those in the aerospace sector, which require expensive tooling. By optimizing the use of tools, the proposed approach can help reduce production costs, improve production efficiency and maintain competitiveness. When manufacturing small series of products, the proposed approach can help meet customer demand by reducing production while improving tool utilization. The discussion is processed at a good level. Mathematical notations respect the necessary details and accurately express the formulations presented by the authors in the contribution.

The article is prepared at the required level. I have two comments about him:

1. Adjust the quality of images number 8-13, because they are quite unclear.

2. formulate the scientific contribution, because the practical contribution is indisputable.

After editing the post according to the comments, the text can be published.

Author Response

First, we would like to express our gratitude to the Editor for providing us with the opportunity to resubmit a revised version of the paper. We also wish to extend our appreciation to the Reviewers for their thorough examination of the manuscript and for their valuable comments, which have allowed us to significantly enhance the original version of the paper. We have diligently addressed each observation raised by the Reviewers, and the manuscript has been amended accordingly.

Modifications made to the text in response to the Reviewers' comments and suggestions are clearly indicated in the manuscript, with changes to the English form highlighted in red, and significant modifications or additions to the content marked in blue. Below, we provide our response to each specific comment brought to our attention. We hope that we have successfully addressed all concerns and that the paper is now suitable for publication in Journal of Manufacturing and Materials Processing.

Comments No 1: [..] 1. Adjust the quality of images number 8-13, because they are quite unclear”

Our response: Thank you very much for this comment. We apologise for the quality of the results presentation. Following this comment, we have thoroughly revised Figures 8 to 13 to make them consistent in colours and more visible in the results display. We have also revised both the text and the image captions.

Comments No 2: “formulate the scientific contribution, because the practical contribution is indisputable”

Our response: We greatly appreciate your comment and sincerely apologise for not being clear in formally outlining the scientific contribution of our work. To address this, we have made significant revisions to the final part of the Introduction, clearly identifying the scientific contribution of the present work by points and appropriately deriving the gap with respect to the current state of the literature. You can find this addition highlighted in blue.

Round 2

Reviewer 1 Report

This paper proposes a genetic algorithm approach to solve the Identical Parallel Machines Problem with Tooling Constraints in Job Shop Flexible Manufacturing Systems with the consideration of tool wear. The approach takes into account the residual useful life of tools and allocates a set of jobs with specific processing times and tooling requirements on identical parallel machines. After the previous reviews and modifications, the quality of the revised paper has been greatly improved. Overall, it is recommended to accept.

Reviewer 2 Report

I have read the paper once more. In this revision, it could perfectly propose a genetic algorithm approach to solve the Identical Parallel Machines Problem with Tooling Constraints in Job Shop Flexible Manufacturing Systems.

So, it can be accepted as it is.